# Prevalence of Involuntary Environmental Cannabis and Tobacco Smoke Exposure in Multi-Unit Housing

**DOI:** 10.3390/ijerph16183332

**Published:** 2019-09-10

**Authors:** Alanna K. Chu, Pamela Kaufman, Michael Chaiton

**Affiliations:** Ontario Tobacco Research Unit, Dalla Lana School of Public Health, University of Toronto, Toronto, ON M5T 3M7, Canada

**Keywords:** cannabis, tobacco, secondhand smoke, multi-unit housing

## Abstract

No research has examined the prevalence of involuntary cannabis exposure in the home within the context of multi-unit housing (MUH). The 2017 cycle of the Centre for Addiction and Mental Health Monitor population RDD survey included measures of environmental cannabis smoke (ECS) and environmental tobacco smoke (ETS) for Ontario, Canada. These ECS measures were defined for those who did not live in a detached dwelling self-reporting noticing any tobacco or cannabis smoke enter the home from a neighboring unit or from outside the building at least once in the past 6 months. Overall, 6.6% (95% CI: 4.5–9.5%) and 7.5% (9% CI: 5.4–10.4%) of the population reported being exposed to ETS and ECS in MUH respectively. Individuals exposed to ECS were single, had used cannabis in the past 12 months, and had lower household incomes. The prevalence of involuntary exposure to cannabis smoke is similar to exposure to tobacco smoke. Exposure correlates were primarily associated with characteristics of those who lived in MUH who tend to be members of more vulnerable populations.

## 1. Introduction

Multi-unit housing (MUH) is predominant in many urban regions in North America and includes semi-detached homes, townhouses, apartment buildings and condominiums. Approximately 34% of Canadians resided in MUH in 2016 [1], and this proportion is expected to rise with the price of single-family homes, declining household size, pressures of long commutes, increasing immigration, an aging population, and land shortages [2]. MUH is unique in that different households share the same indoor air which means that the behaviors in one household have the potential to directly impact other households in the building. This is of particular concern when occupant behavior such as smoking tobacco, cannabis or other products impacts the experience and health of other residents [3]. It has been well established that tobacco smoke can transfer from one unit to another through walls, ductwork, windows and ventilation systems in MUH resulting in involuntary exposure to environmental tobacco smoke (ETS) [4,5,6]. The transfer of environmental cannabis smoke (ECS) in MUH has been less studied; however, given that the contaminant profiles of ETS and ECS are similar [7], it is reasonable to expect cannabis smoke to transfer just as easily. A study of secondhand smoke exposure among MUH residents in California found that some participants perceived the transfer of ECS into their unit as a nuisance and health hazard [8].

Exposure to ETS is associated with adverse health effects, including causal associations with lung cancer, stroke, coronary heart disease, low birth weight infants, and nasal irritation [9]. Although there is a dearth of high quality research on the health effects of ECS, findings conclude that ECS contains many of the same toxic chemicals in ETS that are known to cause heart disease and respiratory illness [7,10], and ECS is associated with significant impairment of blood vessel function in animal studies [11]. ECS has also been shown to produce detectable urine and blood cannabinoid levels, impaired cognitive performance, and sedative drug effects in non-ventilated condition [8,12]. While not specific to a MUH environment, a California study of hospitalized children with a parent enrolled in a tobacco cessation program found that about half of the children showed evidence of urine cotinine and marijuana metabolites suggesting involuntary exposure to tobacco and cannabis [13].

The predictors of ETS exposure is well documented. Sociodemographic factors such as younger age, lower levels of educational attainment, and lower income are associated with ETS [14]. Furthermore, a study found that within MUH, younger age, being female, being non-white, and having a lower household income were associated with higher odds of exposure to ETS [15]. However, little is known about the factors associated with exposure to ECS.

Given the growing literature on the negative health effects of ECS and the legalization of recreational cannabis in Canada on 17 October 2018, it is important to identify baseline measures for the prevalence of ECS in MUH, as well as factors which may increase the risk of exposure. These data are needed to inform evidence-based policies to address environmental smoke exposures in MUH. Most existing policies on environmental smoke have focused on regulations for multi-unit dwellings, rather than exposure due to individuals smoking within the home. Thus, the purpose of this study is to determine the prevalence of ECS and ETS from outside of the home in Ontario prior to legalization and identify the associated sociodemographic factors.

## 2. Materials and Methods

### 2.1. CAMH Monitor

Our study utilizes data collected for the 2017 cycle of the Centre for Addiction and Mental Health (CAMH) Monitor. The CAMH Monitor is a cross-sectional, computer assisted telephone survey which uses a two-stage probability sampling (telephone number from regional stratum) of Ontario adults (≥18 years old) conducted from January to December 2017. The survey used dual-frame random digit dialing (RDD; 90% landline and 10% cell phone numbers). The cooperation rate of the entire sample was 46% and the response rate was 35%. Participants were randomized to complete one of two versions (Panel A or B) of the survey. Only participants on panel A received questions about ETS and ECS.

### 2.2. Measures

All variables were self-reported. Demographic variables included age category (18–34, 35–54, 55+), race (white vs. not white), education status (less than high school vs. more than high school), marital status (married, previously married, or never married), employment status (employed vs. not employed), household income (<$80,000 or ≥$80,000), location (rural vs. non-rural), recent immigration (within the last 20 years), self-reported mental health (fair-poor vs. excellent-good), self-report physical health (fair-poor vs. excellent-good), primary dwelling (detached single family home, attached house, multiple unit dwelling (e.g., apartment), shared accommodation), past 12 month cannabis use, and smoking status (current, former, or never more than 100 cigarettes).

The outcome variables for ETS and ECS were defined as having noticed any tobacco or cannabis smoke enter the home from a neighboring unit or from outside the building at least once in the past 6 months (yes/no). The outcome variables (ETS and ECS) were only asked to participants who responded that their primary dwelling was an MUH including an attached house, multiple-unit dwelling, or shared accommodation.

### 2.3. Analysis

A total sample of 2812 respondents completed the survey. Our analysis is based on a subpopulation of individuals who were in Panel A (n = 999) who responded to the primary dwelling question (n = 992). Data was weighted using the post-stratification sample scaled dual frame weights to provide population number estimates. Demographic data was calculated for the analytic sample as well as prevalence of ETS and ECS.

Bivariate associations of ETS and ECS with gender, age group, education, marital status, employment status, household income, location, recent immigration, self-reported mental and physical health, past 12 month cannabis use and smoking status was calculated using chi-squared tests.

A logistic regression analysis was conducted to obtain the odds ratios of demographic variables on ETS and ECS. A similar logistic regression was conducted within individuals who live in MUH (i.e. attached home, multiple unit dwelling or shared accommodation). All analyses were conducted using Stata 15.0 software (StataCorp. 2017. Stata Statistical Software: Release 15. College Station, TX, USA: StataCorp LLC) accounting for the complex survey design.

## 3. Results

Less than 25% of the population reported that their primary dwelling type was MUH, including approximately 14.8% (95% CI: 11.8–18.3%) in attached homes and 7.1% (95%CI: 5.5–9.1%) in multiple unit dwellings (e.g., apartments). Overall, 6.6% (95% CI: 4.5–9.5%) and 7.5% (95% CI: 5.4–10.4%) of the population reported being exposed to ETS and ECS respectively. This equates to approximately 728,000 individuals exposed to ETS (95% CI: 447,000–1,009,000) and 827,000 individuals exposed to ECS (95% CI: 540,000–1,115,000) in Ontario. Approximately 470,000 individuals reported being exposed to both ETS and ECS (95% CI: 239,000–701,000).

Individuals exposed to ETS in MUH were significantly younger than those not exposed (Table 1). A significantly higher proportion of individuals exposed had a higher education and had used cannabis in the past 12 months compared to those not exposed. Similarly, a significant proportion of individuals exposed to ECS in MUH were younger and a larger proportion had used cannabis in the past 12 months (Table 2). A larger proportion of exposed individuals were current tobacco smokers and reported poor mental health compared to individuals who were not exposed to ECS. Exposed and not exposed individuals also differed significantly on marital status, where a larger proportion of not exposed individuals were married or partnered (64.2% vs. 39.5%).

Adults age 35–54 (aOR: 0.15, 95%CI: 0.03, 0.63, *p* = 0.010) and older adults 55+ (aOR: 0.05, 95% CI: 0.01, 0.22, *p* < 0.001) were less likely to be exposed to ETS compared to younger adults in MUHs (Table 3). Being female (aOR: 2.61, 95%CI: 1.04, 6.54, *p* = 0.04) and previously married (aOR: 2.53, 95% CI: 1.01, 6.36, *p* = 0.048) were associated with higher odds of being exposed to ETS, whereas being never married was associated with lower odds (aOR: 0.14, 95% CI: 0.03, 0.67, *p* = 0.014). Lastly, the odds of ETS was 4.76 times (95% CI: 1.65, 13.71, *p* = 0.004) higher amongst who used cannabis within the past 12 month compared to those who has not.

Within individuals living in MUH, only the association between ETS exposure and older age (55+, aOR: 0.09, 95% CI: 0.02, 0.48, *p* = 0.002), gender (aOR: 4.19, 95% CI: 1.32, 13.37, *p* = 0.015), and never being married (aOR: 0.04, 95%CI: 0.01, 0.26, *p* = 0.001) remained significant. Additionally, the positive association between past 12 month cannabis smoking and ETS exposure became stronger (aOR: 6.68, 95% CI: 1.68, 26.64, *p* = 0.007).Interestingly the significant association between past 12 month cannabis use diminished, while a significantly higher odds of exposure was found in never smokers (never 100 cigarettes; aOR: 8.66, 95% CI: 2.06, 36.32, *p* = 0.003).

Unlike ETS, age was not a significant predictor of ECS exposure (Table 4). Having been previously married (aOR: 2.38, 95% CI: 1.04, 5.46, *p* = 0.041) and having smoked cannabis in the past 12 months (aOR: 3.36, 95% CI: 1.42, 7.94, *p* = 0.006) was associated with higher odds of exposure to ECS. Having a household income of $80,000 or more was associated with lower odds (aOR: 0.33, 95% CI: 0.14, 0.78, *p* = 0.012). Within individuals living in MUH, ECS was not significantly associated with any demographic factors.

## 4. Discussion

In Ontario, 7.5% of the population reported being exposed to cannabis smoke in their residences prior to the legalization of recreational cannabis in Canada in 2018. Similarly, 6.6% of Ontarians reported exposure to tobacco smoke over the same time period. These results provide an understanding of the demographic factors that put individuals at higher risk of exposure to ETS and ECS. Without taking housing type into account, we found that individuals exposed to ECS differed significantly from those not exposed in terms of marital status, household income, past 12 month cannabis use, and never use of cigarettes. However, within MUH, these demographic factors were no longer predictive of ECS exposure, indicating that the predictors of exposure are those associated with living in MUH. These results suggest that all individuals living in MUH are at risk of ECS exposure, regardless of demographic factors. It follows that the primary point of intervention should be policies aimed at reducing cannabis smoking inside MUH. This can be accomplished through targeted education campaigns to raise awareness of the health risks of exposure; and by encouraging and supporting the adoption of smoke-free housing policies at municipal and provincial levels. Smoke-free policies in MUH may also support cessation behavior, including reducing smoking and increasing quit attempts among residents who smoke [16,17].

Past year cannabis use was strongly associated with ECS within MUH. It is possible that individuals who are cannabis smokers may be better able to detect ECS compared to individuals who are not familiar with its odor. It is also possible that there is a clustering of cannabis users, whereby individuals who smoke cannabis are more likely to live in MUH where other cannabis users live. The variances between those reporting exposure to ECS and ETS in MUH suggests that there may be a systematic difference in the individuals who are at risk of ECS or ETS—either in the factors which cause individuals to self-report exposure or in the type of MUH in which they reside. Past year cannabis users also had a higher odds of reporting both ETS and ECS, but never smokers only had a higher odds of reporting ETS. It is possible that past year tobacco users are more able to identify cannabis as distinct from tobacco smoke, whereas never smokers are unable to detect the difference between tobacco and cannabis use due to a lack of experience. These hypotheses should be explored in more detail in future research but suggest that self-report measures may underestimate the extent of exposure to both ETC and ECS.

Interestingly, the demographic factors related to ECS were different than those associated with ETS. Within MUH, older age and being male was significantly associated with lower odds of ETS, but not with ECS. A similar study of self-reported involuntary ETS exposure in MUH also found lower odds of exposure amongst males and older individuals [15], Moreover, having never been married was strongly associated with ETS, but not with ECS.

This study is subject to a number of limitations. The CAMH Monitor uses RDD, which inherently excludes individuals without a phone number. There may be a systematic difference in the individuals who are choosing to report, thus this study may be subject to response bias. This measure only asks about ETS and ECS from adjacent neighboring units or outside of the building and does not include other important exposure environments such as within the household, at work, or outside the home.

This study uses the definition of involuntary exposure to ETS and ECS from sources outside the home (e.g., from a neighboring unit) because this is relevant to public health and important for the development of smoke-free policy regulations in MUH. The issue of secondhand exposure from smoking that occurs inside the home is an important source of exposure for children and other individuals living in the home. Consequently, the estimates here underestimate the total burden of environmental smoke. The focus of this paper is with the narrow definition used for regulatory policy. Studies that measure air quality and cannabis smoking behavior in different environments and dwelling types will increase our understanding of exposure. Measuring exposures in children living in MUH is also an important avenue of research. With the growing use of cannabis, monitoring the prevalence of ECS and ETS in MUH is needed to fully understand the extent of involuntary exposures within an evolving policy and regulation landscape.

## 5. Conclusions

This study provides new information on the associations between sociodemographic factors and environmental cannabis and tobacco smoke exposures in MUH. Future research at the local, provincial and national level is necessary to better understand the prevalence and burden associated with ETS and ECS in MUH but these data show that exposure to cannabis smoke is a widespread issue for many people and requires public health and regulatory consideration.

## Figures and Tables

**Table 1 ijerph-16-03332-t001:** Prevalence of Exposure to Environmental Tobacco Smoke, Ontario, Canada, 2017.

Variable	Not Exposed		Exposed		
	N	% (95%CI)	N	% (95%CI)	*p*-Value
**Age Category**					<0.001
18–34	112	23.3 (19.3,27.7)	15	52.2 (33.8,70.1) ^†^	
35–54	261	34.2 (30.3,38.4)	17	32.5 (18.6,50.3) ^†^	
55+	564	42.5 (38.5,46.6)	18	15.3 (8.4,26.3) ^†^	
**Gender**					
Male	412	52.3 (48.1,56.5)	17	37.8 (22.2,56.5) ^†^	0.170
Female	527	47.7 (43.5,51.9)	33	62.2 (43.5,77.8)	
**Race**					0.163
Non-White	83	14.4 (11.4,18.0)	10	26.4 (11.1,50.8) ^††^	
White	842	85.6 (82.0,88.6)	38	73.6 (49.2,88.9)	
**Education**					0.003
High school or less	276	25.3 (21.8,29.1)	6	7.4 (2.8,18.1) ^††^	
More than high school ^1^	650	74.7 (70.9,78.2)	44	92.6 (81.9,97.2)	
**Marital Status**					0.124
Married/partner	602	63.6 (59.2,67.7)	23	45.8 (28.3,64.5) ^†^	
Previously married	197	11.5 (9.5,13.8)	14	24.2 (9.9,48.4) ^††^	
Never married	136	25.0 (21.0,29.5)	12	29.9 (15.4,50.0) ^†^	
**Employment status ^2^**					0.972
Not employed	486	39.8 (35.9,43.9)	19	40.2 (22.5,60.8) ^†^	
Employed	447	60.2 (56.1,64.1)	30	59.8 (39.2,77.5) ^†^	
**Household Income**					0.545
<$80,000	355	40.5 (36.0,45.2)	26	47.0 (27.8,67.2) ^†^	
$80,000+	372	59.5 (54.8,64.0)	16	53.0 (32.8,72.2) ^†^	
Location					0.282
Non-Rural	800	88.5 (85.9,90.6)	46	93.6 (82.0,97.9)	
Rural	139	11.5 (9.4,14.1)	4	6.4 (2.1,18.0) ^††^	
**Recent Immigrant (past 20 years)**					0.145
no	906	95.0 (92.5,96.7)	45	89.0 (73.0,96.1)	
yes	26	5.0 (3.3,7.5) ^†^	5	11.0 (3.9,27.0) ^††^	
**Overall Health**					0.682
Excellent–Good	804	87.3 (84.4,89.8)	46	83.5 (53.6,95.7)	
Fair–Poor	132	12.7 (10.2,15.6)	4	16.5 (4.3,46.4) ^††^	
**Overall Mental Health**					0.166
Excellent–Good	852	90.2 (87.3,92.4)	43	79.2 (52.3,93.0)	
Fair–Poor	80	9.8 (7.6,12.7)	6	20.8 (7.0,47.7) ^††^	
**Main Residence**					<0.001
Detached single family home	770	82.1 (78.6,85.1)			
Attached home	93	11.1 (8.6,14.1)	28	67.3 (50.0,81.0)	
Multiple unit dwelling	66	5.4 (4.0,7.2) ^†^	20	29.9 (17.0,46.9) ^†^	
Shared accommodation	10	1.5 (0.7,3.4) ^††^	2	2.8 (0.5,13.1) ^††^	
**Past 12 month Cannabis use**					0.001
Yes	135	19.2 (15.9,23.0)	15	48.4 (29.8,67.5) ^†^	
No	793	80.8 (77.0,84.1)	34	51.6 (32.5,70.2) ^†^	
**Smoking Status**					0.452
current	134	16.4 (13.3,20.0)	6	23.0 (8.5,49.1) ^††^	
Former	329	28.2 (24.8,31.9)	14	18.1 (9.3,32.4) ^†^	
Never 100 cigarettes	471	55.4 (51.2,59.6)	30	58.9 (38.7,76.5) ^†^	

^†^ Moderately unstable (16.6%–33.2%). ^††^ Highly unstable (>33.2%). ^1^ Includes some post-secondary education and above. ^2^ Employed includes full-time, part-time and self-employed, not employed includes unemployed, homemaker, retired, student and other.

**Table 2 ijerph-16-03332-t002:** Prevalence of Exposure to Environmental Cannabis Smoke, Ontario, Canada, 2017.

Variable	Not Exposed		Exposed		
	N	% (95%CI)	N	% (95%CI)	*p*-Value
**Age Category**					0.028
18–34	111	23.7 (19.7,28.3)	16	42.4 (26.0,60.7) ^†^	
35–54	260	34.1 (30.1,38.3)	18	34.2 (20.6,50.9) ^†^	
55+	562	42.2 (38.2,46.2)	23	23.4 (12.8,38.7) ^†^	
**Gender**					0.531
Male	404	50.9 [46.7,55.2]	27	56.8 [39.0,73.0]	
Female	531	49.1 [44.8,53.3]	30	43.2 [27.0,61.0] †	
**Race**					0.169
White	838	85.7 (82.1,88.6)	45	74.7 (52.3,88.8) ^††^	
Non-White	83	14.3 (11.4,17.9)	10	25.3 (11.2,47.7)	
**Education**					0.663
High school or less	271	24.3 (21.0,28.1)	12	21.0 (10.2,38.2) ^††^	
More than high school ^1^	651	75.7 (71.9,79.0)	45	79.0 (61.8,89.8)	
**Marital Status**					0.012
Married/partner	605	64.2 (59.8,68.4)	21	39.5 (24.6,56.5) ^†^	
Previously married	195	11.2 (9.3,13.5)	17	26.0 (12.3,46.9) ^††^	
Never married	131	24.6 (20.5,29.1)	18	34.5 (20.6,51.7) ^†^	
**Employment status ^2^**					0.294
Not employed	478	39.2 (35.2,43.2)	30	48.6 (31.9,65.6) ^†^	
Employed	450	60.8 (56.8,64.8)	27	51.4 (34.4,68.1) ^†^	
**Household Income**					0.061
<$80,000	349	39.4 (34.9,44.1)	34	58.2 (38.8,75.4) ^†^	
$80,000+	372	60.6 (55.9,65.1)	17	41.8 (24.6,61.2) ^†^	
**Location**					0.149
Non-Rural	795	88.3 (85.8,90.4)	54	94.9 (84.4,98.5)	
Rural	140	11.7 (9.6,14.2)	3	5.1 (1.5,15.6)^††^	
**Recent Immigrant (past 20 years)**					0.254
no	901	94.9 (92.4,96.7)	54	90.6 (76.9,96.6)	
yes	27	5.1 (3.3,7.6) ^†^	3	9.4 (3.4,23.1) ^††^	
**Overall Health**					0.104
Excellent–Good	808	87.9 (85.0,90.3)	44	76.2 (54.3,89.6)	
Fair–Poor	124	12.1 (9.7,15.0)	13	23.8 (10.4,45.7) ^††^	
**Overall Mental Health**					0.001
Excellent–Good	852	91.0 (88.3,93.1)	45	69.5 (47.7,85.1)	
Fair–Poor	77	9.0 (6.9,11.7)	10	30.5 (14.9,52.3) ^†^	
**Main Residence**					<0.001
Detached single family home	770	82.8 (79.3,85.8)			
Attached home	95	11.6 (9.0,14.9)	27	53.3 (36.4,69.4)	
Multiple unit dwelling	62	4.8 (3.5,6.5)	26	35.5 (22.2,51.6) ^†^	
Shared accommodation	8	0.8 (0.3,2.1) ^††^	4	11.2 (3.5,30.2) ^††^	
**Past 12 month Cannabis use**					<0.001
Yes	126	18.5 (15.1,22.4)	24	53.0 (36.1,69.2) ^†^	
No	797	81.5 (77.6,84.9)	33	47.0 (30.8,63.9)	
**Smoking Status**					0.003
Current	124	15.1 (12.1,18.7)	17	38.1 (22.1,57.2) ^†^	
Former	328	28.1 (24.7,31.8)	16	20.7 (11.3,34.7) ^†^	
Never 100 cigarettes	478	56.8 (52.6,60.9)	24	41.2 (26.1,58.2) ^†^	

^†^ Moderately unstable (16.6%–33.2%). ^††^ Highly unstable (>33.2%). ^1^ Includes some post-secondary education and above. ^2^ Employed includes full-time, part-time and self-employed, not employed includes unemployed, homemaker, retired, student and other.

**Table 3 ijerph-16-03332-t003:** Odds Ratios of Exposure to Environmental Tobacco Smoke Versus Non-exposed Individuals Living in Multi-Unit Housing, Ontario, Canada, 2017.

Variable	Total Sample (n = 992)		Within MUH (n = 229)	
	OR (95%CI)	*p*-Value	OR (95%CI)	*p*-Value
**Age Category**				
18–34	Ref		Ref	
35–54	0.15 (0.03, 0.63)	0.010	0.24 (0.04,1.53)	0.132
55+	0.05 (0.01, 0.22)	<0.001	0.09 (0.02,0.48)	0.005
**Gender**				
Male	Ref		Ref	
Female	2.61 (1.04, 6.54)	0.040	4.19 (1.32,13.37)	0.015
**Race**				
Non-White	Ref		Ref	
White	0.70 (0.22, 2.24)	0.550	0.90 (0.27,3.04)	0.866
**Employment status^2^**				
No	Ref		Ref	
Yes	0.65 (0.26, 1.64)	0.362	0.59 (0.18,1.85)	0.362
**Education**				
High school or less	Ref		Ref	
More than high school ^1^	3.31 (0.83, 13.28)	0.091	3.53 (0.80,15.48)	0.095
**Marital Status**				
Married/partner	Ref		Ref	
Previously married	2.53 (1.01, 6.36)	0.048	0.62 (0.18,2.05)	0.430
Never married	0.14 (0.03,0.67)	0.014	0.04 (0.01,0.26)	0.001
**Household Income**				
<$80,000	Ref		Ref	
$80,000+	0.48 (0.17,1.39)	0.176	1.00 (0.35,2.84)	0.994
**Location**				
Non-Rural	Ref		Ref	
Rural	0.40 (0.10,1.51)	0.174	0.79 (0.10,6.19)	0.824
**Recent Immigrant (past 20 years)**				
no	Ref		Ref	
yes	1.43 (0.22,9.24)	0.707	0.36 (0.07,1.93)	0.236
**Overall Health**				
Excellent-Good	Ref		Ref	
Fair-Poor	0.74 (0.16,3.47)	0.703	0.31 (0.05,1.86)	0.200
**Overall Mental Health**				
Excellent–Good	Ref		Ref	
Fair–Poor	0.65 (0.16,2.65)	0.549	1.80 (0.34,9.63)	0.489
**Past 12 month Cannabis use**				
No	Ref		Ref	
Yes	4.76 (1.65,13.71)	0.004	6.68 (1.68,26.64)	0.007
**Smoking Status**				
Current	Ref		Ref	
Former	1.21 (0.41,3.56)	0.728	1.66 (0.32,8.52)	0.543
Never 100 cigarettes	2.22 (0.81,6.10)	0.120	8.66 (2.06,36.32)	0.003

^1^ Includes some post-secondary education and above. ^2.^ Employed includes full-time, part-time and self-employed, not employed includes unemployed, homemaker, retired, student and other.

**Table 4 ijerph-16-03332-t004:** Odds Ratios of Exposure to Environmental Cannabis Smoke Versus Non-exposed Individuals Living in Multi-Unit Housing, Ontario, Canada, 2017.

Variable	Total Sample (n = 992)		Within MUH (n = 229)	
	OR (95%CI)	*p*-Value	OR (95%CI)	*p*-Value
**Age Category**				
18–34	Ref		Ref	
35–54	0.80 (0.16,4.10)	0.790	2.53 (0.61,10.52)	0.203
55+	0.47 (0.06,3.66)	0.473	0.81 (0.17,3.79)	0.790
**Gender**				
Male	Ref		Ref	
Female	0.87 (0.43,1.77)	0.703	0.66 (0.22,1.94)	0.449
**Race**				
Non-White	Ref		Ref	
White	0.71 (0.28,1.79)	0.466	1.05 (0.36,3.09)	0.928
**Employment status^2^**				
No	Ref		Ref	
Yes	0.83 (0.32,2.11)	0.687	0.35 (0.12,1.05)	0.060
**Education**				
High school or less	Ref		Ref	
More than high school^1^	1.13 (0.40,3.20)	0.823	1.15 (0.34,3.82)	0.824
**Marital Status**				
Married/partner	Ref		Ref	
Previously married	2.38 (1.04,5.46)	0.041	0.80 (0.28,2.30)	0.680
Never married	0.99 (0.17,5.86)	0.992	1.11 (0.28,4.45)	0.886
**Household Income**				
<$80,000	Ref		Ref	
$80,000+	0.33 (0.14, 0.78)	0.012	0.97 (0.33,2.85)	0.957
**Location**				
Non-Rural	Ref		Ref	
Rural	0.30 (0.05,1.70)	0.172	1.30 (0.12,14.40)	0.833
**Recent Immigrant (past 20 years)**				
no	Ref		Ref	
yes	0.94 (0.17,5.11)	0.947	0.23 (0.04,1.29)	0.095
**Overall Health**				
Excellent–Good	Ref		Ref	
Fair–Poor	0.62 (0.20,1.91)	0.405	0.48 (0.12,1.95)	0.305
**Overall Mental Health**				
Excellent–Good	Ref		Ref	
Fair–Poor	2.96 (0.71,12.39)	0.138	4.44 (0.84,23.45)	0.079
**Past 12 month Cannabis use**				
No	Ref		Ref	
Yes	3.36 (1.42,7.94)	0.006	3.28 (0.95,11.33)	0.060
**Smoking Status**				
Current	Ref		Ref	
Former	0.47 (0.18,1.26)	0.135	0.43 (0.12,1.58)	0.202
Never 100 cigarettes	0.45 (0.20, 1.02)	0.056	0.54 (0.16,1.84)	0.321

^1^ Includes some post-secondary education and above. ^2^ Employed includes full-time, part-time and self-employed, not employed includes unemployed, homemaker, retired, student and other.

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
