# Peer review of "Prevalence of Involuntary Environmental Cannabis and Tobacco Smoke Exposure in Multi-Unit Housing"

_ijerph, 2019, doi:10.3390/ijerph16183332_

Round 1
Reviewer 1 Report
The authors present data from a large monitor study. Although it is unclear whether they present data from 2812 respondents or several millions based on the claims they make in the discussion section. Also, some elaboration on the interpretation of the ORs and a comparison between residents of MUH and the total sample are lacking. Nevertheless, the paper is generally well-written and concise. Please find my comments, listed per section of the paper, below.
Abstract:
ECS and ETS abbreviations are not written out at first use as usual, but at second use
Introduction:
Please provide a few examples of the negative health effects of ECS with references in similar populations (MUH).
Materials and Methods:
The question about tobacco/cannabis smoke contains the word “any”. Does this mean that also a single, spurious exposure over 6 months’ time was weighed the same as a frequent exposure from for example a neighboring unit? If yes, please consider rephrasing your results to “individuals ever exposed to ETS/ECS”
Results:
Regarding the claim to how many individuals were exposed in Ontario; please consider taking uncertainty into account like you do with the 95% Cis. So for example not “approximately 1,296,000 individuals” but “between 1,000,000 and 1,300,000 individuals”. Stating one approximate number provides a false sense of certainty as they are extrapolated from a sample of 2812 respondents. From the tables presenting ORs it is not evident what the reference group was. In the title it only says “Odds ratios of Cannabis ..”. Please consider rephrasing to “Odds ratios for exposure to Cannabis Second Hand Smoke versus non-exposed” or something similar. Please consider indicating the number of participants in the total sample and within MUH to put the 95% CIs into perspective (also for Table 1 & 2, exposed/not exposed) Please explain that a protective OR in this case means that some who is older (for example) is less likely to be exposed to ECS compared to younger MUH residents. Just mentioning that the odds increase or decrease might get a bit confusing for the average reader without some elaboration on this.
Discussion:
The first lines of the discussion are incorrect. If I understand your methods section correctly, you did not get data from millions of people so you cannot state that over 1 million people reported being exposed to cannabis smoke. You can report the percentage in your study with 2812 respondents though. Moreover, the rates in your study might be overestimated by response bias as you already mentioned, so I would strongly discourage using the “million” figures in any communication about this study. If you did get data on this from millions of people, please indicate so in the first paragraph of the methods section.
Conclusions:
Please delete “nevertheless”. The conclusion should be its own statement and not follow from a previous sentence.

Author Response
The authors present data from a large monitor study. Although it is unclear whether they present data from 2812 respondents or several millions based on the claims they make in the discussion section. Also, some elaboration on the interpretation of the ORs and a comparison between residents of MUH and the total sample are lacking. Nevertheless, the paper is generally well-written and concise. Please find my comments, listed per section of the paper, below.
Abstract:
Comment #1: ECS and ETS abbreviations are not written out at first use as usual, but at second use
Response: This has been corrected (Line 11-12).
Introduction:
Comment #2: Please provide a few examples of the negative health effects of ECS with references in similar populations (MUH).
Response: There are no data that we are aware of that examine the health effects of ECS among residents of MUH. Therefore, we have included literature on general health effects of ECS and highlighted the potential comparability.
Materials and Methods:
Comment #3:The question about tobacco/cannabis smoke contains the word “any”. Does this mean that also a single, spurious exposure over 6 months’ time was weighed the same as a frequent exposure from for example a neighboring unit? If yes, please consider rephrasing your results to “individuals ever exposed to ETS/ECS”
Response: In the methods, we have clarified that our definition of ETS and ECS includes having noticed any tobacco or cannabis smoke enter the home from a neighbouring unit or from outside the building at least once in the past 6 months.
Results:
Comment #4: Regarding the claim to how many individuals were exposed in Ontario; please consider taking uncertainty into account like you do with the 95% Cis. So for example not “approximately 1,296,000 individuals” but “between 1,000,000 and 1,300,000 individuals”. Stating one approximate number provides a false sense of certainty as they are extrapolated from a sample of 2812 respondents.
Response: We have now included confidence intervals for the population estimates. Thank you for pointing out this issue, as we checked the weighting and made a correction to the population estimates using the appropriate population expansion weighting for the panels.
Comment #5: From the tables presenting ORs it is not evident what the reference group was. In the title it only says “Odds ratios of Cannabis ..”. Please consider rephrasing to “Odds ratios for exposure to Cannabis Second Hand Smoke versus non-exposed” or something similar.
Response: We have changed this for clarity.
“Table 3. Odds Ratios for exposure to Environmental Tobacco Smoke versus non-exposed individuals living in Multi-Unit Housing, Ontario, Canada, 2017”
“Table 4. Odds Ratios of exposure to Environmental Cannabis Smoke versus non-exposed individuals living in Multi-Unit Housing, Ontario, Canada, 2017”
Comment #6: Please consider indicating the number of participants in the total sample and within MUH to put the 95% CIs into perspective (also for Table 1 & 2, exposed/not exposed)
Response: Yes, the number of participants in the total sample and within MUHs have been added to Table 3 and 4. The number of participants is listed in Table 1 and 2 for exposed and not exposed.
Comment #7: Please explain that a protective OR in this case means that some who is older (for example) is less likely to be exposed to ECS compared to younger MUH residents. Just mentioning that the odds increase or decrease might get a bit confusing for the average reader without some elaboration on this.
Response: These statements have been amended to make the ORs more clear (line 119-140).
Discussion:
Comment #8: The first lines of the discussion are incorrect. If I understand your methods section correctly, you did not get data from millions of people so you cannot state that over 1 million people reported being exposed to cannabis smoke. You can report the percentage in your study with 2812 respondents though. Moreover, the rates in your study might be overestimated by response bias as you already mentioned, so I would strongly discourage using the “million” figures in any communication about this study. If you did get data on this from millions of people, please indicate so in the first paragraph of the methods section.
Response: We have rewritten the first lines of the discussion to highlight the prevalence of exposure rather than the survey population estimate.
Conclusions:
Comment #9: Please delete “nevertheless”. The conclusion should be its own statement and not follow from a previous sentence.
We have deleted “nevertheless” from the first line of the conclusions paragraph (line 216).
Reviewer 2 Report
This is an interesting paper examine the cannabis exposure in Multi- unit housing in Canada.
However, I have several concerns about this manuscript.
The title of this paper is about cannabis secondhand smoke exposure. However, contents include both the environmental cannabis smoke and environmental tobacco smoke in MUH. Therefore, the title and content are not consistent. The definition for ETS and ECS are different from other definition. Therefore it is not proper to make a comparison between this study and others as shown in line 169-173. The authors should explain why they chose to use such definition. According to the definition of the ETS and ECS in this paper, I think it is meaningful only among the participants who do not smoke (or use cannabis themselves) and having a smoke(cannabis) free policy at home. However, it seems such element was not considered in this paper. In the background part, it is not clarified why this paper pay attention to ETS r ECS from neighbor home. In 2018, Canada has legalized the use of cannabis. How does this affect the exposure of E CS exposure at home? What should do for public health practitioners? Is it particularly important to advocate the smoke free policy in MUH? Discussion part should cover such questions.Author Response
This is an interesting paper examine the cannabis exposure in Multi- unit housing in Canada.
However, I have several concerns about this manuscript.
Comment #10: The title of this paper is about cannabis secondhand smoke exposure. However, contents include both the environmental cannabis smoke and environmental tobacco smoke in MUH. Therefore, the title and content are not consistent.
Response: We have changed the title to be more consistent with the content of the study.
Comment #11: The definition for ETS and ECS are different from other definition. Therefore it is not proper to make a comparison between this study and others as shown in line 169-173.
Response: Yes, the reviewer is correct that the references cited use self-report measures of ETS exposure in the home by someone who smokes in the home; they do not focus on MUH environments where ETS can enter the home from another location in the building. For this reason, we removed the references and have amended the title to better reflect the study.
Comment #12: The authors should explain why they chose to use such definition. According to the definition of the ETS and ECS in this paper, I think it is meaningful only among the participants who do not smoke (or use cannabis themselves) and having a smoke (cannabis) free policy at home. However, it seems such element was not considered in this paper. In the background part, it is not clarified why this paper pay attention to ETS r ECS from neighbor home.
Response: We chose to use the definition of involuntary exposure to ETS and ECS from sources outside the home (e.g., from a neighbouring unit) because this is relevant to public health and important for the development of smoke-free policy regulations in MUH. The current regulations on secondhand smoke inside the home deals with the issue of smoke entering from outside the home, particularly in MUHs; this separate from the issue of secondhand smoking from within the house for children and infants. The focus of this paper is with the narrow definition used for regulatory policy. We have added some of this rationale to the end of the introduction and discussed the limitations of our definition in the discussion.
Comment #13: In 2018, Canada has legalized the use of cannabis. How does this affect the exposure of ECS exposure at home? What should do for public health practitioners? Is it particularly important to advocate the smoke free policy in MUH? Discussion part should cover such questions.
Response: Our study analyzed data collected from January to December 2017, thus we cannot address exposure to ECS since cannabis was legalized in Canada in October 2018. However, we do intend to analyze post-legalization data as it becomes available in a future study. The current study provides a baseline for future comparisons that can inform questions about how legalization affects ECS exposure in the home, the role of public health practitioners and the importance of advocating for smoke-free policies in MUH. We have clarified the legalization date in the introduction and discussion.

Reviewer 3 Report
This paper presents interesting and timely data, regarding the demographics and predictors of environmental cannabis smoke exposure. The topic is important, and the paper presents some insights. However it could benefit from some clarification and some potential further discussion.
Specific comments follow.
Abstract:
Line 9: Spell out ECS and ETS first (it is spelled out 4 lines later) Line 13 “reporting” should be “reported.” Please spell out “MURB” (or use MUH, as you use throughout the paper). Line 16 – 17 – the last sentence is awkward and could be rephrased.Introduction:
Consider referring to this important study which showed ECS as defined by urine THC among infants hospitalized for bronchiolitis in the PICU setting - https://www.ncbi.nlm.nih.gov/pubmed/30455340Materials and Methods
P 2 line 57: I would specify that “environmental smoke exposure” is ETS and ECS, assuming it is both Line 69: “The outcome variables (Cannabis ETS and ECS SHS) “– is not consistent with the way the terms are used throughout. Why not just say (ETS and ECS)
Also – if these questions regarding ETS and ECS were only asked of those participants who lived in MUH (if I understand this correctly) then I assume all the subsequent results refer to those within MUH. However, the results discussed on p 3 are confusing. For example – the second paragraph starting on line 100 - - states that the odds of exposure to ETS was significantly lower in adults ages 35 – 54…. This is worded as if it were in the overall sample, especially when comparing it to the next paragraph, starting lin3 106 – “Within individuals living in MUH…”
Results
See comment above, about p 3 being confusing with regard to the whole sample vs. those asked about MUH. This comes up in both the 2ndand 3rdparagraphs, which discuss “within individuals living in MUH” as if this is in comparison to the whole sample. Second paragraph (p 3, line 104 – 106) – it is interesting that having used cannabis within the past 12 months increased odds of exposure to ETS by 5.76 times. Is this also a function of the demographics of those living in MUH? The interaction between one’s neighbors’ tobacco smoking and your own cannabis use is hard to explain otherwise. Titles for tables: To be consistent with the terminology used in the text, I would refer to “environmental tobacco smoke” and “environmental cannabis smoke” rather than “second hand smoke”. Also, should the title of Table 1 read,Prevalence…..in MUH?” Tables 3 and 4 titles are confusing – I think they should be retitled – they are called “Odds Ratios of Cannabis second-hand smoke” but they seem to compare the demographic data between the overall sample (those in non-MUH who were NOT asked the ECS and ETS questions) and those who were asked those questions, who live in MUH. The titles as they are, suggest that you compare the reported ECS and ETS between the overall sample and those in MUH but that is not the case I think.
Discussion
P 7 line 172 – 174 - missing a word – “which” (in between “previous research” and “has found”) Line 175: I would change the first sentence of that paragraph to delete the word “strengths.” This paragraph appropriately discusses limitations You mention the limitation of other exposure environments such as within the household. This is a big limitation which should merit a little more discussion. Furthermore, the finding that past year cannabis use was strongly associated with ECS suggests not only that those who use cannabis themselves are better able to detect ECS, but perhaps they are also more likely to have other cannabis users within their own living space, which also provides ECS. One other limitation which is important is the potential for exposure of children – which is not mentioned – I assume the survey does not ask about this. However, as up to 60% of children in MUH are exposed to ETS it would be important to add the potential for ECS to children, from within their own living space and adjacent units. Again, see the reference I suggest, above. Another point for discussion is that bans on smoking in MUH may decrease the prevalence of smoking, not just the ETS. Recent bans on MUH in the US have not been well studied, but the potential is there.Conclusions
Delete the first word, “nevertheless.” I would think the prevalence and burden associated with ETS is more well understood than that of ECS in MUH -Author Response
This paper presents interesting and timely data, regarding the demographics and predictors of environmental cannabis smoke exposure. The topic is important, and the paper presents some insights. However it could benefit from some clarification and some potential further discussion.
Specific comments follow.
Abstract:
Comment #14: Line 9: Spell out ECS and ETS first (it is spelled out 4 lines later)
Response: This has been corrected (line 11-12).
Comment #15: Line 13 “reporting” should be “reported.”
Response: This has been corrected (line 15).
Comment #16: Please spell out “MURB” (or use MUH, as you use throughout the paper). Line 16 – 17 – the last sentence is awkward and could be rephrased.
Response: This has been corrected (line 10 & 16).
Introduction:
Comment #17: Consider referring to this important study which showed ECS as defined by urine THC among infants hospitalized for bronchiolitis in the PICU setting - https://www.ncbi.nlm.nih.gov/pubmed/30455340
Response: Thank you for this reference - we have cited it in the introduction (line 50-53).
Materials and Methods
Comment #18: P 2 line 57: I would specify that “environmental smoke exposure” is ETS and ECS, assuming it is both Line 69: “The outcome variables (Cannabis ETS and ECS SHS) “– is not consistent with the way the terms are used throughout. Why not just say (ETS and ECS)
Response: Apologies for these mistakes. They have been corrected (line 74 & 86).
Comment #19: Also – if these questions regarding ETS and ECS were only asked of those participants who lived in MUH (if I understand this correctly) then I assume all the subsequent results refer to those within MUH. However, the results discussed on p 3 are confusing. For example – the second paragraph starting on line 100 - - states that the odds of exposure to ETS was significantly lower in adults ages 35 – 54…. This is worded as if it were in the overall sample, especially when comparing it to the next paragraph, starting lin3 106 – “Within individuals living in MUH…”
Response: You are correct that only individuals who lived in MUHs were asked about ETS and ECS (due to ETS and ECS referring to smoke entering the home from a neighboring unit or from outside the building). However, we examined the prevalence of exposure amongst the general population (including those not in MUHs, n = 992) in Tables 1 and 2, discussed in line 111-118. We also examined the associations (ORs) between demographic factors and exposure in the overall sample (n = 992, line 119-127) and within only MUHs (n = 229) in Tables 3 and 4 respectively (line 128-135). Sample sizes have been added to Table 3 and 4 to make clear the difference between analyses.
Results
Comment #19: See comment above, about p 3 being confusing with regard to the whole sample vs. those asked about MUH. This comes up in both the 2ndand 3rdparagraphs, which discuss “within individuals living in MUH” as if this is in comparison to the whole sample. Second paragraph (p 3, line 104 – 106) – it is interesting that having used cannabis within the past 12 months increased odds of exposure to ETS by 5.76 times. Is this also a function of the demographics of those living in MUH? The interaction between one’s neighbors’ tobacco smoking and your own cannabis use is hard to explain otherwise.
Response: See above response for clarification regarding the difference between “within MUHs” and the total sample. We examined the associations between demographic factors and exposure for both the total sample and within the MUHs. The association between past 12 month cannabis use and exposure to ETS appears both within the total sample (aOR 4.76) and within only MUHs (aOR 6.68).
The focus of this study is not the correlates of demographics and living in MUH, as this has been the subject of prior research. Rather, we are interested in the target population of these policies to address ECS and ETS in MUH. For cannabis, it appears that exposure is not related to these demographic factors (Table 4), rather it is overwhelmingly about living in MUHs, as stated in the discussion.
Comment #20: Titles for tables: To be consistent with the terminology used in the text, I would refer to “environmental tobacco smoke” and “environmental cannabis smoke” rather than “second hand smoke”. Also, should the title of Table 1 read, Prevalence…..in MUH?” Tables 3 and 4 titles are confusing – I think they should be retitled – they are called “Odds Ratios of Cannabis second-hand smoke” but they seem to compare the demographic data between the overall sample (those in non-MUH who were NOT asked the ECS and ETS questions) and those who were asked those questions, who live in MUH. The titles as they are, suggest that you compare the reported ECS and ETS between the overall sample and those in MUH but that is not the case I think.
Response: The titles of the Tables have been changed to use consistent terminology (i.e. environmental tobacco smoke and environmental cannabis smoke).
Table 1 and 2 refer to prevalence of ETS and ECS within the sample that completed Panel A (n=992) of the CAMH Monitor, not only those who live in MUHs. Therefore, the not exposed group include individuals who do not live in MUHs and do live in MUHs but have not experienced ETS or ECS, and the exposed group include only individuals who live in MUHs and have been exposed to ETS or ECS respectively.
Table 3 and 4 describe the odds ratios of exposure to ETS and ECS compared to non-exposed individuals living in MUHs, respectively. Each table describes two sets of ORs: (1) The ORs of ETS or ECS in exposed vs. non-exposed within the total sample (n=992, all those asked the question), and (2) The ORs of ETS and ECS in exposed vs. non-exposed within only the individuals who reported living in MUHs (n=229). The titles of the tables have been changed to better describe the data, and the sample sizes have been added for clarity.
Discussion
Comment #21: P 7 line 172 – 174 - missing a word – “which” (in between “previous research” and “has found”) Line 175: I would change the first sentence of that paragraph to delete the word “strengths.” This paragraph appropriately discusses limitations You mention the limitation of other exposure environments such as within the household. This is a big limitation which should merit a little more discussion.
Response: The word “which” has been added (line 202) and the word “strengths” has been deleted (line 204). We agree that the exposure to ETS and ECS within the household is important. However, there currently exists literature on self-reported exposure to tobacco within the household. The focus of this study is specifically on smoke that enters from others’ homes within multi-unit housing, as this exposure is amenable to policy change. We have included further discussion on this limitation.
"This study uses the definition of involuntary exposure to ETS and ECS from sources outside the home (e.g., from a neighbouring unit) because this is relevant to public health and important for the development of smoke-free policy regulations in MUH. The issue of secondhand exposure from smoking that occurs inside the home is an important source of exposure for children and other individuals living in the home. Consequently, the estimates here underestimate the total burden of environmental smoke. The focus of this paper is with the narrow definition used for regulatory policy. Studies that measure air quality and cannabis smoking behaviour in different environments and dwelling types will increase our understanding of exposure. Measuring exposures in children living in MUH is also an important avenue of research. With the growing use of cannabis, monitoring the prevalence of ECS and ETS in MUH is needed to fully understand the extent of involuntary exposures within an evolving policy and regulation landscape"
Comment #22: Furthermore, the finding that past year cannabis use was strongly associated with ECS suggests not only that those who use cannabis themselves are better able to detect ECS, but perhaps they are also more likely to have other cannabis users within their own living space, which also provides ECS.
Response: Agreed that this may be the case – however the data only captures ETS and ECS from outside of the home or from other units and we have clarified that our sample is limited as it does not include sources of secondhand smoke from within the home.

Round 2
Reviewer 2 Report
It seems that the authors have answered my questions in a rational way. I'm fine with the paper.
Reviewer 3 Report
Thank you for addressing my comments. I feel this revised version is significantly clearer and makes a stronger point about the need for further research and legislation. I have only a few very minor comments:
P 2 lines 43 – 48 -- Ref. 13 is a Colorado study, not California P2 line 49 – predictors “are” not is P 2 line 64 I would keep tenses consistent – “Our study utilized data” (not “utilizes”) P3 line 94: Should be “were” not “was: (“Bivariate associations …were….) P3 line 119 missing a d - - “Associated:” not “associate” P 3 line 121 – 122 fix grammar: missing “those” before the line” who used cannabis within the past month…” and “has” should be “had” at end of that sentence. P 3 line 133 should be “were” not “was” (“having been previously married and having smoked cannibis were…”)